# DIFFERENCE-OF-SUBMODULAR BREGMAN DIVERGENCE

**Masanari Kimura**[*]
School of Mathematics and Statistics
The University of Melbourne
Victoria, Australia
m.kimura@unimelb.edu.au

**Takahio Kawashima**[*]
ZOZO Research
ZOZO Next, Inc.
Chiba, Japan
takahiro.kawashima@zozo.com

**Tasuku Soma & Hideitsu Hino**
Institute of Statistical Mathematics / RIKEN AIP
Tokyo, Japan
{soma, hino}@ism.ac.jp

## ABSTRACT

The Bregman divergence, which is generated from a convex function, is commonly used as a pseudo-distance for comparing vectors or functions in continuous spaces. In contrast, defining an analog of the Bregman divergence for discrete spaces is nontrivial. Iyer & Bilmes (2012b) considered Bregman divergences on discrete domains using submodular functions as generating functions, the discrete analogs of convex functions. In this paper, we further generalize this framework to cases where the generating function is neither submodular nor supermodular, thus increasing the flexibility and representational capacity of the resulting divergence, which we term the difference-of-submodular Bregman divergence. Additionally, we introduce a learnable form of this divergence using permutation-invariant neural networks (NNs) and demonstrate through experiments that it effectively captures key structural properties in discrete data. As a result, the proposed method significantly improves the performance of existing methods on tasks such as clustering and set retrieval problems. This work addresses the challenge of defining meaningful divergences in discrete settings and provides a new tool for tasks requiring structure-preserving distance measures.

## 1 INTRODUCTION

A divergence is a principal concept that determines the geometric structure of a space of interest, which is formally defined as follows:

**Definition 1.1.** Let $\Omega$ be a set. A function $D : \Omega \times \Omega \to \mathbb{R}$ is called a divergence on $\Omega$ if $D$ satisfies the following conditions: for all $x, y \in \Omega$,

$$D(x, y) \geq 0, \text{ and}$$
$$D(x, y) = 0 \iff x = y.$$

The Bregman divergence is a class of divergences that measure dissimilarity between two points based on a strictly convex function. Given a strictly convex and differentiable function $f$, the Bregman divergence $D_f$ between two points $\mathbf{x}$ and $\mathbf{y}$ in a convex set $\Omega \subseteq \mathbb{R}^d$ is defined as:

$$D_f(\mathbf{x}, \mathbf{y}) = f(\mathbf{x}) - f(\mathbf{y}) - \langle \nabla f(\mathbf{y}), \mathbf{x} - \mathbf{y} \rangle \tag{1}$$

where $\nabla f(\mathbf{y})$ is the gradient of $f$ evaluated at $\mathbf{y}$ and $\langle \cdot, \cdot \rangle$ denotes the inner product. It generalizes the squared Euclidean distance and the Kullback–Leibler divergence (Kullback & Leibler, 1951). Typically, divergences are defined on differential manifolds, such as a Euclidean space and a family

---

[*]Equal contribution.

of distributions. Compared to these continuous spaces, it is more challenging to define a reasonable and capable metric on discrete spaces, such as $2^V$ with $V = \{1, \ldots, N\}$ being a finite ground set. Although there are numerous choices for measuring a distance or similarity between subsets $X, Y \in 2^V$, most known metrics in the literature are constructed by counting the sizes of $X \cap Y, X \setminus Y$, and other simple set operations (Choi et al., 2010). Such classical metrics easily become less meaningful when the ground set $V$ is large because rarely do same elements co-occur in two subsets of $V$.

The *submodular-Bregman divergence* (Iyer & Bilmes, 2012b) is the first to tackle this problem with the theory of submodular set functions. It is based on the fact that subgradients and supergradients can be defined for any submodular function. Therefore, a Bregman-like divergence can be defined through a submodular function $f$ in the way analogous to the standard Bregman divergence (1).

Although the submodular-Bregman divergence is quite natural and intuitive, it is not clear whether it satisfies the definition of divergences. This problem comes from *identifiability* of the divergence: $D(x, y) = 0 \implies x = y$ in Definition 1.1. In the usual Bregman divergence (1), the strict convexity is required for $f$ to guarantee the identifiability. Similar to the usual Bregman divergence, the subgradients or supergradients are insufficient to make it a divergence; it may be necessary to assume the existence of strict subgradients or strict supergradients.

Another issue in the submodular-Bregman divergence is that the choice of the submodular function is ad-hoc. Although Iyer & Bilmes (2012b) introduces several concrete examples of the submodular function, the resulting divergences are again the forms with respect to simple set operations. So it is unclear whether they actually overcome the difficulty of the classical metrics between sets. Therefore, a more flexible framework for handling the submodular-Bregman divergence is necessary, and it would be even better if we could extend the capability of submodular-Bregman divergence.

**Our contribution.** In this paper, we introduce a novel class of divergences on discrete spaces that is strictly more expressive than submodular Bregman divergences and propose a learning framework of the divergences through permutation-invariant NNs. First, we formally show that the submodular-Bregman divergence is indeed a divergence if $f$ is strictly submodular. By extending this observation, we further propose a new class of divergences that can be defined even if $f$ is neither submodular nor supermodular. Since the concrete way to compose the divergence depends on the difference-of-submodular decomposition (Narasimhan & Bilmes, 2005; Iyer & Bilmes, 2012b; Li & Du, 2020), we call it *difference-of-submodular Bregman divergences*. We show that the expressive power of difference-of-submodular Bregman divergences defined by set function $f$ gets richer if the class of the underlying set function $f$ gets larger. We then propose a learnable form of difference-of-submodular Bregman divergences based on submodular permutation-invariant NNs. Numerical experiment shows that learnable difference-of-submodular Bregman divergences can capture the crucial structure and significantly improves the performance of existing methods in downstream tasks.

**Related work.** Bregman divergences play a central role in various areas of statistics and machine learning, including clustering, regression, and optimization. For example, the classical $k$-means algorithm is a special case of Bregman $k$-means clustering, where the squared Euclidean distance is replaced with a Bregman divergence. This generalization allows for clustering based on different divergence measures, providing more flexibility (Banerjee et al., 2005). The Bregman divergence is closely related to the exponential family of distributions and generalized linear models. The divergence can be derived from the convex conjugate of the log-partition function of the exponential family (Wainwright & Jordan, 2008). In information geometry, it is used to define the geometry of statistical models and provides a framework for understanding various divergence measures (Amari, 2016). The Bregman divergence is also appeared in matrix factorization techniques, such as nonnegative matrix factorization with a Bregman divergence, which allows different divergence measures to be used in the factorization process (Cichocki & Amari, 2010). In optimization, it is used in mirror descent algorithms and other first-order optimization methods. These methods benefit from the divergence's properties to achieve better convergence (Beck & Teboulle, 2003). In variational inference, the Bregman divergence measures the difference between the true posterior distribution and the variational approximation (Blei et al., 2017). Faust et al. (2023) elaborates the theory of difference-of-convex algorithms through the lens of the Bregman divergence. By considering Bregman divergences on a discrete space, it is expected that the wealth of knowledge provided by these studies can be utilized.

Distance metric learning (DML (Ye et al., 2018; Wang & Sun, 2014)) is a technique used to learn a distance metric that can effectively measure the similarity or dissimilarity between instances. This is crucial for improving the performance of various machine learning tasks such as classification (Weinberger et al., 2005; Davis et al., 2007; Goldberger et al., 2005), clustering (Xing et al., 2002; Kulis et al., 2005; Ye et al., 2007), and information retrieval (Song et al., 2016; Schroff et al., 2015; Wang et al., 2014). Min et al. (2009) and Salakhutdinov & Hinton (2007) used NNs to learn representations that make metric learning easier. There is also research in distance metric learning that aims to learn Bregman divergences from data (Li et al., 2023; Rezaei et al., 2023; Lu et al., 2023; Siahkamari et al., 2020). However, to the best of our knowledge, there is no research that learns Bregman divergences over discrete sets.

## 2 PRELIMINARIES

In this section, we introduce submodular and supermodular functions and their differentials, the concept of strict modularity, and permutation-invariant NNs necessary for the flexible realization of submodular and supermodular functions, as the building blocks of the divergence proposed in this paper.

### 2.1 SUBMODULAR FUNCTIONS AND SEMIDIFFERENTIALS

**Definition 2.1.** A set function $f : 2^V \to \mathbb{R}$ is said to be submodular if

$$f(X) + f(Y) \geq f(X \cup Y) + f(X \cap Y)$$

holds for every $X, Y \subseteq V$. In addition, $f$ is supermodular if $-f$ is submodular, and $f$ is modular if $f$ is both submodular and supermodular.

A modular function $m : 2^V \to \mathbb{R}$ is known to be written by $m(X) = \sum_{i \in X} m(i)$, and it is identified as the vector $(m_1, \ldots, m_N) := (m(1), \ldots, m(N)) \in \mathbb{R}^N$ (we define $m(\emptyset) := 0$). Thus, the inner product of two modular functions $m, m' : 2^V \to \mathbb{R}$ is calculated as the inner product of the two vectors: $\langle m, m' \rangle := \sum_{i=1}^N m(i)m'(i) = \sum_{i=1}^N m_i m'_i$.

Submodular functions are often understood as discrete versions of convex functions. Indeed, subdifferentials can be defined for a submodular function $f$ (Fujishige, 2005). Interestingly, superdifferentials are also defined for a submodular function (Iyer & Bilmes, 2012b):

**Definition 2.2.** Let $f : 2^V \to \mathbb{R}$ be a submodular function. The set of modular functions

$$\partial_f(Y) := \{m \in \mathbb{R}^N : \forall X \subseteq V, \ m(X) - m(Y) \leq f(X) - f(Y)\}$$

is called the subdifferential of $f$ at $Y \subseteq V$, and $h_Y \in \partial_f(Y)$ is called a subgradient of $f$ at $Y \subseteq V$. Similarly, the set of modular functions

$$\partial^f(Y) := \{m \in \mathbb{R}^N : \forall X \subseteq V, \ m(X) - m(Y) \geq f(X) - f(Y)\}$$

is called the superdifferential of $f$ at $Y \subseteq V$, and $g_Y \in \partial^f(Y)$ is called a supergradient of $f$ at $Y \subseteq V$.

Sub- and super-differentials are together referred to as semidifferentials (Iyer et al., 2013). In general, there are infinitely many choices to pick a subgradient $h_Y \in \partial_f(Y)$ (see the proof of Proposition A.4) and the extreme point can be obtained by a greedy algorithm (Edmonds, 1970). On the other hand, Iyer & Bilmes (2012b) proposes three types of supergradients $\hat{g}_Y, \check{g}_Y, \bar{g}_Y \in \partial^f(Y)$ and they are called *grow*, *shrink*, and *bar* supergradients, respectively (Iyer et al., 2013). Table 1 shows the form of these supergradients.

### 2.2 STRICT SUBMODULARITY

Recall that Bregman divergences are defined using strictly convex functions. We analogously need to introduce strict submodularity for set functions.

**Definition 2.3.** A set function $f : 2^V \to \mathbb{R}$ is said to be strictly submodular if

$$f(X) + f(Y) > f(X \cup Y) + f(X \cap Y) \tag{2}$$

Table 1: Supergradients proposed in (Iyer & Bilmes, 2012b). We denote $f(j|Y) := f(\{j\} \cup Y) - f(Y)$ for any $Y \subseteq V$ and $j \in V$.

|  | $j \in Y$ | $j \notin Y$ |
|---|---|---|
| $\hat{g}_Y(j)$ | $f(j|V\backslash\{j\})$ | $f(j|Y)$ |
| $\check{g}_Y(j)$ | $f(j|Y\backslash\{j\})$ | $f(j|\emptyset)$ |
| $\bar{g}_Y(j)$ | $f(j|V\backslash\{j\})$ | $f(j|\emptyset)$ |

for every non-comparable[1] $X, Y \subseteq V$ and $f$ is strictly supermodular if $-f$ is strictly submodular.

For example, the facility location function is an important class of submodular functions. Let $\phi_{ij}$ be non-negative values for $i \in V = \{1, \ldots, N\}$ and $k = 1, \ldots, K$. Then, the facility location function is written as $f_{\mathrm{FC}} : X \mapsto \sum_{k=1}^{K} \max_{i \in X} \phi_{ik}$. More generally, the log-sum-exp relaxation of the facility location function,

$$f_{\mathrm{FC},\varepsilon}(X) := \varepsilon \sum_{k=1}^{K} \log \sum_{i \in X} e^{\phi_{ik}/\varepsilon}, \tag{3}$$

is strictly submodular for $\varepsilon > 0$ (see Appendix A.2 for the proof) and it converges to $f_{\mathrm{FC}}$ as $\varepsilon \to 0$.

For a strictly submodular function, we can define strict semidifferentials:

**Definition 2.4.** Let $f : 2^V \to \mathbb{R}$ be a strictly submodular function. The set of modular functions

$$\tilde{\partial}_f(Y) := \{m \in \mathbb{R}^N : \forall X \in 2^V \backslash \{Y\}, \ m(X) - m(Y) < f(X) - f(Y)\} \tag{4}$$

is called the strict subdifferential of $f$ at $Y \subseteq V$, and $\tilde{h}_Y \in \tilde{\partial}_f(Y)$ is called a strict subgradient of $f$ at $Y \subseteq V$. Similarly

$$\tilde{\partial}^f(Y) := \{m \in \mathbb{R}^N : \forall X \in 2^V \backslash \{Y\}, \ m(X) - m(Y) > f(X) - f(Y)\} \tag{5}$$

is called the strict superdifferential of $f$ at $Y \subseteq V$, and $\tilde{g}_Y \in \tilde{\partial}^f(Y)$ is called a strict supergradient of $f$ at $Y \subseteq V$.

The non-emptiness of $\tilde{\partial}_f$ is shown in Appendix A.1. All of the three supergradients $\hat{g}_Y, \check{g}_Y$, and $\bar{g}_Y$ defined in Table 1 satisfy the definition of the strict supergradients if $f$ is strictly supermodular.

**Proposition 2.5** (Strict supergradients). *The modular functions $\hat{g}_Y, \check{g}_Y$, and $\bar{g}_Y$ defined in Table 1 are all the strict supergradients if $f$ is strictly supermodular.*

See Appendix A.1 for the proof.

## 2.3 PERMUTATION-INVARIANT NEURAL NETWORKS

Permutation invariance is the key concept in leveraging modern NN architectures to model set-structured data (Zaheer et al., 2017). Let $V = \{1, \ldots, N\}$ be the index set and $\mathcal{X}_N = \{[\mathbf{x}_1, \ldots, \mathbf{x}_M] \in \mathbb{R}^{D \times M} : \mathbf{x}_1, \ldots, \mathbf{x}_M \in \mathbb{R}^D, \ M \leq N\}$ be the entire set of matrices with $N$ or fewer columns. We consider an NN $f_{\mathrm{NN}} : \mathcal{X}_N \to \mathcal{Y}$, where $\mathcal{Y}$ is the output space[2]. The permutation invariance requests $f_{\mathrm{NN}}$ to behave as a set function on $2^V$.

**Definition 2.6.** The function $f_{\mathrm{NN}}$ is said to be permutation invariant if it satisfies

$$f_{\mathrm{NN}}([\mathbf{x}_1, \ldots, \mathbf{x}_M]) = f_{\mathrm{NN}}([\mathbf{x}_{\pi_M(1)}, \ldots, \mathbf{x}_{\pi_M(M)}])$$

for any $M \in V$, any inputs $\mathbf{x}_1, \ldots, \mathbf{x}_M$, and any permutation $\pi_M : \{1, \ldots, M\} \to \{1, \ldots M\}$.

---

[1] $X$ and $Y$ are said to be non-comparable if neither $X \subseteq Y$ nor $Y \subseteq X$.

[2] The elements of the input $\mathbf{x}_1, \ldots, \mathbf{x}_M$ are not necessarily real vectors, but we consider real vectors for simplicity.

The permutation invariance and permutation invariant NN are first introduced along with Deep Sets (Zaheer et al., 2017). Today, many effective permutation-invariant architectures are developed such as PointNet (Qi et al., 2017), Set Transformer (Lee et al., 2019), Perceiver (Jaegle et al., 2021), and SetVAE (Kim et al., 2021). See (Kimura et al., 2024; Xie & Tong, 2025) for comprehensive reviews about permutation-invariant NNs.

In particular, PointNet is convenient for modeling submodular functions. As in (3), we define the PointNet architecture in the generalized form with log-sum-exp.

**Definition 2.7.** Let $\phi := [\phi_1, \ldots, \phi_K] : \mathbb{R}^D \to \mathbb{R}^K$, $\gamma : \mathbb{R}^K \to \mathbb{R}$, $\varepsilon \geq 0$, and $X \subseteq V$. When $\varepsilon > 0$, the architecture of $\varepsilon$-PointNet $f_{\mathrm{PN},\varepsilon}$ is written by

$$f_{\mathrm{PN},\varepsilon}([\mathbf{x}_i]_{i \in X}) = \gamma\left(\left(\varepsilon \log \sum_{i \in X} e^{\phi_1(\mathbf{x}_i)/\varepsilon}, \ldots, \varepsilon \log \sum_{i \in X} e^{\phi_K(\mathbf{x}_i)/\varepsilon}\right)^\top\right). \tag{6}$$

For $\varepsilon = 0$, it is defined as

$$f_{\mathrm{PN},0}([\mathbf{x}_i]_{i \in X}) = \gamma\left(\left(\max_{i \in X} \phi_1(\mathbf{x}_i), \ldots, \max_{i \in X} \phi_K(\mathbf{x}_i)\right)^\top\right).$$

In $\varepsilon$-PointNet, the log-sum-exp or max operations realize permutation invariance. For $\varepsilon > 0$, $f_{\mathrm{PN},\varepsilon}$ reduces to the vanilla PointNet (i.e., $f_{\mathrm{PN},0}$) with the $\varepsilon \to 0$ if $\gamma$ is continuous. Since $\phi$ and $\gamma$ are arbitrary functions, we can design them using NNs freely. Given $\mathbf{x}_1, \ldots, \mathbf{x}_N$, $f_{\mathrm{PN},\varepsilon}([\mathbf{x}_i]_{i \in X})$ is regarded as the set function $f_{\mathrm{PN},\varepsilon}(X) := f_{\mathrm{PN},\varepsilon}([\mathbf{x}_i]_{i \in X})$. Especially, if $\varepsilon > 0$, $\phi_1, \ldots, \phi_K$ are all non-negative function, and $\gamma$ is the summation function (namely, $\gamma(\mathbf{x}) = \sum_i x_i$), $\varepsilon$-PointNet $f_{\mathrm{PN},\varepsilon}$ reduces to the relaxed facility location function (3), which leads to strict submodularity.

## 3 THEORETICAL FOUNDATIONS

### 3.1 SUBMODULAR-BREGMAN DIVERGENCES

The generalized Bregman divergence is the variant of the Bregman divergence (1) in which the gradient is replaced by a subgradient. Iyer & Bilmes (2012b) introduced submodular-Bregman divergences as a subclass of the generalized Bregman divergence. The earlier work, however, did not explicitly mention that in order for the identifiability of the divergence to always hold, it is necessary for the submodular functions involved to be strict. Complementarily, we consider how the submodular-Bregman divergences become *proper*, satisfying Definition 1.1. Hereafter, we denote $1_X$ as the indicator vector of a set $X \subseteq V$.

**Theorem 3.1.** *Let $f : 2^V \to \mathbb{R}$ be a strictly submodular function. For every $Y \subseteq V$, there exist modular functions $h_Y, g_Y \in \mathbb{R}^N$ that make*

$$D_f(X, Y) := f(X) - f(Y) - \langle h_Y, 1_X - 1_Y \rangle, \tag{7}$$

*and*

$$D^f(X, Y) := -f(X) + f(Y) + \langle g_Y, 1_X - 1_Y \rangle \tag{8}$$

*satisfy the conditions of divergences shown in Definition 1.1, respectively.*

*Proof.* For every $Y \subseteq V$, we can take a strict subgradient of $f$ at $Y$ for $h_Y$, i.e., $h_Y \in \tilde{\partial}_f(Y)$. This choice of the modular function $h_Y$ leads

$$D_f(X, Y) = f(X) - f(Y) - h_Y(X) + h_Y(Y) \geq f(X) - f(Y) - f(X) + f(Y) = 0$$

for all $X \subseteq V$ and the equality holds if and only if $X = Y$. This $D_f$ satisfies the conditions in Definition 1.1. We can also show that $D^f$ satisfies Definition 1.1 with a strict supergradient $g_Y \in \tilde{\partial}^f(Y)$ in the same manner. □

In Iyer & Bilmes (2012b), $D_f$ and $D^f$ are referred to as lower-bound submodular-Bregman divergence and upper-bound submodular-Bregman divergence, respectively. The submodular-Bregman divergence $D_f$ depends on the (strictly) subgradient map $\mathcal{H}_f : Y \mapsto \tilde{h}_Y \in \tilde{\partial}_f(Y)$ since the choice of a (strict) subgradient in $D_f$ is not unique. Although the dependency should be clarified as $D_f^{\mathcal{H}_f}$, we omit it for simplicity (do the same for $D^f$).

## 3.2 Discrete Bregman Divergences

Intuitively, the expressive power of $f$ may directly affect the expressive power of the submodular-Bregman divergences (7), (8) (in fact, this is true and will be proved in Subsection 3.3). If we can extend the function class that can be taken as $f$, the flexibility of the divergence may increase. Surprisingly, the submodular-Bregman divergences (7), (8) are well-defined even if $f$ is a general set function not necessarily submodular (nor supermodular). The strong DS (difference-of-submodular) decomposition (Li & Du, 2020) is the key idea.

**Theorem 3.2** (Strong DS Decomposition (Li & Du, 2020)). *Every set function $f : 2^V \to \mathbb{R}$ can be decomposed into the difference of two monotone increasing and strictly submodular functions $f^1$ and $f^2$, i.e., $f = f^1 - f^2$.*

Note that a set function $f^1 : 2^V \to \mathbb{R}$ is said to be monotone increasing when $f^1(Y) > f^1(X)$ for any $X \subset Y \subseteq V$. The (weak) DS decomposition is introduced in (Narasimhan & Bilmes, 2005) and proved in (Iyer & Bilmes, 2012a) rigorously and combinatorially. The proof of strong DS decomposition provided in (Li & Du, 2020) is constructed by tweaking that in (Iyer & Bilmes, 2012a).

Once the strong DS decomposition $f = f^1 - f^2$ is obtained, the strict subgradients of $f$ at $Y \subseteq V$ can be constructed by $h_Y = h_Y^1 - g_Y^2 \in \tilde{\partial}_f(Y)$ with $h_Y^1 \in \tilde{\partial}_{f^1}(Y), g_Y^2 \in \tilde{\partial}^{f^2}(Y)$, even if $f$ is neither strictly submodular nor supermodular, and the same is true for strict supergradients $g_Y \in \tilde{\partial}^f(Y)$. This fact enables us to consider *submodular-Bregman divergences with non-submodular set functions*.

**Theorem 3.1'.** *For every set function $f : 2^V \to \mathbb{R}$ and every $Y \subseteq V$, there exist modular functions $h_Y, g_Y \in \mathbb{R}^N$ that make (7) and (8) satisfy the conditions in Definition 1.1, respectively.*

In this paper, we call the divergences justified by Theorem 3.1' the *difference-of-submodular Bregman divergence* (DBD) since such divergences no longer need the (strict) submodularity of $f$. Generally speaking, finding (strong) DS decomposition $f = f^1 - f^2$ given $f$ takes exponential complexity. One possible approach to benefit from Theorem 3.1' is to prepare the submodular functions $f^1$ and $f^2$ beforehand and to construct non-submodular $f$ (Section 4).

## 3.3 Expressive Power of Discrete Bregman Divergences

We extended the submodular-Bregman divergence to the DBD which bases a non-submodular set function in general. Our question is whether the extension really improves the expressive power as a divergence. More generally, does the expressive power of the DBD $D_f$ increase when a class of the set function $f$ becomes richer? For example, $\varepsilon$-PointNet constitutes a superclass of the relaxed facility location functions as described in Subsection 2.3. Clearly, the class of all submodular functions is a subclass of the class of all DS functions, or all set functions.

For precision, we define a class of set functions on $2^V$ as a set of set functions closed under the addition of modular functions.

**Definition 3.3.** Let $\mathcal{F}$ be a set of set functions on $2^V$. We denote the *class* of the set $\mathcal{F}$ as

$$\mathcal{C}(\mathcal{F}) := \{f + m : f \in \mathcal{F}, m \in \mathbb{R}^N\}$$

That is, $f \in \mathcal{C}(\mathcal{F})$ implies $f + m \in \mathcal{C}(\mathcal{F})$ for any modular function $m \in \mathbb{R}^N$. For example, the set of modular, facility location, submodular, PointNet, and DS functions all induce classes of the corresponding sets. We denote the set of all DBDs of set functions in $\mathcal{C}(\mathcal{F})$ by $\mathcal{D}_{\mathcal{C}(\mathcal{F})}$.

**Theorem 3.4.** *Let $\mathcal{F}, \mathcal{F}'$ be sets of set functions and $\mathcal{C} := \mathcal{C}(\mathcal{F}), \mathcal{C}' := \mathcal{C}(\mathcal{F}')$ be the classes induced by $\mathcal{F}$ and $\mathcal{F}'$, respectively. If $\mathcal{C} \subset \mathcal{C}'$, then $\mathcal{D}_{\mathcal{C}} \subset \mathcal{D}_{\mathcal{C}'}$.*

*Proof.* Take a set function $f' \in \mathcal{C}' \backslash \mathcal{C}$. Suppose for the contrary that a DBD with respect to $f'$ can be represented by that of another set function $f \in \mathcal{C}$. From $D_{f'}(X, Y) = D_f(X, Y)$ for all $X, Y \subseteq V$, we have $D_{f'}(X, \emptyset) = D_f(X, \emptyset)$ for all $X \subseteq V$. Nevertheless, $D_{f'}(X, \emptyset)$ is the sum of $f'(X)$ and a modular function, and the same for $D_f(X, \emptyset)$. Hence $f'(X)$ can be written as the sum of a set function in $\mathcal{C}$ (namely $f$) and a modular function, which contradicts $f' \in \mathcal{C}' \backslash \mathcal{C}$. $\qquad\square$

Theorem 3.4 claims that the extension of the submodular-Bregman divergence to the DBD actually increases the expressive power. This motivates us to consider non-submodular set functions to obtain more capable divergences.

## 4 PROPOSED METHOD

Following Theorem 3.1$'$ and Theorem 3.4, we propose a capable and learnable divergence that can adapt downstream tasks. Although Theorem 3.2 states the existence of the (strong) DS decomposition such that $f = f^1 - f^2$ for any $f : 2^V \to \mathbb{R}$, the good method to find (strictly) submodular functions $f^1$ and $f^2$ is unknown.

To avoid this difficulty, we prepare two submodular NNs *a priori* and take them as $f^1$ and $f^2$. Given $Y \subseteq V$, we can find the subgradient of $f^1$, $h_Y^1 \in \partial_{f^1}(Y)$, and the supergradient of $f^2$, $g_Y^2 \in \partial^{f^2}(Y)$. Then, we measure the dissimilarity between $X \subseteq V$ and $Y$ as

$$
\begin{aligned}
D_f(X, Y) &= D_{f^1}(X, Y) + D^{f^2}(X, Y) \\
&= f^1(X) - f^1(Y) - h_Y^1(X) + h_Y^1(Y) - f^2(X) + f^2(Y) + g_Y^2(X) - g_Y^2(Y). \quad (9)
\end{aligned}
$$

In downstream tasks, the DBD (9) can be learned by some objective function of metric learning, such as the triplet loss Hoffer & Ailon (2018).

## 5 NUMERICAL EXPERIMENTS

We consider revealing the behavior of the DBD from numerical experiments. The datasets, network architecture, and other details are as follows.

**Datasets**   In our experiments, we use the following datasets.

- **MNIST** (Deng, 2012): The MNIST dataset is a collection of handwritten digits with a training set of 60,000 examples and a test set of 10,000 examples. Each example is a $28 \times 28$ image, making it a 784-dimensional vector. For an illustrative example (Section 5.1), we use the MNIST dataset to develop a discrete problem setting.
- **ModelNet40** (Wu et al., 2015): The ModelNet40 dataset consists of 12,311 meshes in 40 categories (such as airplane and chair), of which 9,843 are used for training, while the rest 2,468 are reserved for testing. The corresponding point clouds are uniformly sampled from the mesh surfaces. Such point cloud data can be regarded as a set of vectors and is therefore suitable for the discrete problem setting that is the focus of this study. We utilize this dataset to demonstrate the real-world applicability of our DBD (Section 5.2).

**Network Architecture**   The choice of the NN architecture used as a component of the DBD is arbitrary, as long as submodularity is guaranteed. In our implementation, we employ $\varepsilon$-PointNet (6) with $K = 1$, the identity function as $\gamma$, $\varepsilon = 0$ or $0.001$, and a fully-connected multi-layer perceptron (MLP) as $\phi$. For both $f^1$ and $f^2$, the MLPs consist of two hidden layers of 64 units. For the activation functions, the ReLU is used for the hidden and final layers, which yields non-negative outputs; thus the submodularity is guaranteed. In Section 5.2, we also experiment on the DBD without DS decomposition as the ablation study. For fairness, the MLP within the model without decomposition is adjusted to have $64 \times 128$ units in the two hidden layers.

**Learning Procedure**   For learning the DBD, we use the triplet loss (Hoffer & Ailon, 2018):

$$
\mathcal{L}(f) := \sum_{i=1}^{n} \max\left( D_f(X_A^i, X_P^i) - D_f(X_A^i, X_N^i), 0 \right),
$$

where $X_A^i, X_P^i, X_N^i \subseteq V$ are $i$-th anchor, positive, and negative sets, whose construction method will be explained later. We also use Adam (Kingma & Ba, 2015) as the optimization algorithm for the gradient update of the NNs, with a batch size of 64 and a learning rate of $0.001$ unless otherwise stated. The extreme point is taken as the subgradient $h_Y^1$ (Edmonds, 1970) as in (Iyer & Bilmes, 2012b) and the *grow*, *shrink*, and *bar* supergradients are taken as $g_Y^2$.

| Query set | Reference sets | DBD |
|:---:|:---:|:---:|
|  |  | 0.173 |
|  |  | 0.309 |
|  |  | 0.772 |
|  |  | 0.237 |
|  |  | 0.677 |
|  |  | 1.304 |

Figure 1: Illustrative example on MNIST dataset (Deng, 2012). The query and reference sets are constructed from instances in the MNIST dataset, and the DBD values correspond to the divergences between the respective set pairs.

## 5.1 ILLUSTRATIVE EXAMPLE

First, we design a toy experiment to investigate whether our DBD works properly as a divergence. As a divergence function, it is expected to (i) always take non-negative values for any set pair and (ii) take smaller values for similar set pairs and larger values for those that are not. Here, (i) is clearly ensured by Theorem 3.1' and the implementation described in Section 4. In this subsection, we consider a qualitative evaluation using the MNIST dataset to check the behavior of (ii). The original MNIST dataset is a collection of handwritten digit images and we develop a discrete problem setting from this dataset. In this toy experiment, we collect instances in the original dataset to generate $n$ sets of a certain size $M$. The collection of these sets $\mathcal{D}$ is considered as a new discrete version of the dataset, which is used to train the DBD. We set $n = 50,000$, $M = 3, 4$, and learn 200 epochs. Now consider the mapping $y : \mathbf{x} \mapsto y(\mathbf{x})$ from each instance $\mathbf{x} \in \mathbb{R}^{784}$ of the original dataset to its corresponding label $y(\mathbf{x})$, and consider the mapping defined as $\mathcal{Y} : X = \{\mathbf{x}_1, \ldots, \mathbf{x}_M\} \mapsto \mathcal{Y}(X) = \{y(\mathbf{x}_1), \ldots, y(\mathbf{x}_M)\}$. Given the anchor set $X_A^i \in \mathcal{D}$, the positive set $X_P^i$ and the negative set $X_N^i$ are sampled with ensuring the following conditions: $X_P^i \in \left\{ X \in \mathcal{D} : \mathcal{Y}(X_A^i) \cap \mathcal{Y}(X) \neq \emptyset \right\}$, $X_N^i \in \left\{ X \in \mathcal{D} : \mathcal{Y}(X_A^i) \cap \mathcal{Y}(X) = \emptyset \right\}$.

Figure 1 shows the experimental results for this toy example with $\varepsilon = 0$. For each query set, DBD values between the query and some reference sets are reported. Here, the query and reference sets are arbitrarily chosen so that there are differences in the expected divergence values. For example, the top row of each group of reference sets is chosen to be similar to the query set in terms of the labels of the original dataset, while the bottom row is chosen to be far from it. From this result, we can see that the DBD takes smaller values for similar set pairs and larger values for those that are not.

## 5.2 APPLICATIONS ON REAL-WORLD DATASET

Next, we confirm the usefulness of our DBD from numerical experiments on the real-world discrete dataset. To this end, we consider two applications: set clustering and set retrieval on the ModelNet40 dataset. Here, each instance $X \in \mathcal{D}$ of the dataset $\mathcal{D}$ is a point cloud of size $M = 500$ uniformly sampled from the mesh, and a corresponding class label $y(X)$ is given. Since each $X \in \mathcal{D}$ can be regarded as a set of size $M$, the dataset $\mathcal{D}$ satisfies the discrete problem setting we are interested in. Given an anchor set $X_A^i \in \mathcal{D}$, the positive and negative sets $X_P^i \in \left\{ X \in \mathcal{D} : y(X_A^i) = y(X) \right\}$, $X_N^i \in \left\{ X \in \mathcal{D} : y(X_A^i) \neq y(X) \right\}$ are sampled while learning the DBD. In the following, the results of the quantitative evaluation are reported with the means and standard deviations of 10 trials at different random seeds.

Table 2: Experimental results of set clustering for ModelNet40 dataset (Wu et al., 2015). Prefixes *grow*, *shrink*, and *bar* correspond to the supergradients $\hat{g}_Y(j)$, $\check{g}_Y(j)$ and $\bar{g}_Y(j)$ respectively (as shown in Table 1). The means and standard deviations of 10 trials with different random seeds are reported.

| $D_f(X, Y)$ | $f(X)$ | Rand index | |
| --- | --- | --- | --- |
| | | $\varepsilon = 0$ | $\varepsilon = 0.001$ |
| *grow*-DBD w/ decomposition | $f^1(X) - f^2(X)$ | $0.7942(\pm0.0092)$ | $0.8015(\pm0.0060)$ |
| *shrink*-DBD w/ decomposition | $f^1(X) - f^2(X)$ | $0.7905(\pm0.0088)$ | $0.7912(\pm0.0047)$ |
| *bar*-DBD w/ decomposition | $f^1(X) - f^2(X)$ | $0.7410(\pm0.0113)$ | $0.7520(\pm0.0095)$ |
| *grow*-DBD w/o decomposition | $f^1(X)$ | $0.7741(\pm0.0097)$ | $0.7743(\pm0.0092)$ |
| *shrink*-DBD w/o decomposition | $f^1(X)$ | $0.7750(\pm0.0098)$ | $0.7756(\pm0.0080)$ |
| *bar*-DBD w/o decomposition | $f^1(X)$ | $0.7303(\pm0.0125)$ | $0.7325(\pm0.0101)$ |
| $\|X \setminus Y\| + \|Y \setminus X\|$ | $\|X\|$ | $0.0225(\pm0.0060)$ | |
| $1 - \|X \cup Y\|/\|Y\|$ | $1$ | $0.0232(\pm0.0059)$ | |
| $1 - (\|Y\| + \|X \cup Y\|)/2\|Y\|$ | $1/2$ | $0.0232(\pm0.0059)$ | |

**Set Clustering** As one of the downstream tasks in the real-world dataset, we consider set clustering. In this task, we perform clustering with the $k$-means algorithm based on the trained DBD, which is a similar setting to the experiments of Iyer & Bilmes (2012b). Note that the $k$-means algorithm with a Bregman divergence is justified by Banerjee et al. (2005). To investigate the impact of the strict submodularity, which is theoretically required, we conducted experiments with $\varepsilon = 0$ (submodular but not strictly submodular) and $\varepsilon = 0.001$ (strictly submodular), respectively. As an ablation study, the results with a single $\varepsilon$-PointNet as $f(X)$ are reported as w/o decomposition, in which $f^2 = g_Y^2 \equiv 0$ in (9). Also, w/ decomposition corresponds to the DBD based on the DS decomposition.

Table 2 shows the experimental results of the set clustering task. For the performance evaluation, we use the Rand index between the resulting clusters and ground truth labels (e.g., airplaine and chair), where the Rand index is a metric used to evaluate the similarity between two different clusterings of the same dataset. The results with some known special cases of the submodular-Bregman divergence, found in (Iyer & Bilmes, 2012b), are also reported. First, the performance impact of the learnable DBDs is clear because they produce significantly larger Rand index values compared to the submodular Bregman divergences. It can be suggested that this is because the exact intersection tends to be almost zero in many cases, as each element in the set corresponds to the coordinates of a point cloud. In such cases, the high expressive power and learnability of NNs seem to be very effective. Next, we turn our attention to the results of the comparison between the w/ and w/o DS decomposition. These comparisons show that better performance is obtained by the DS decomposition in all three types of supergradients. In addition, looking at the standard deviation of the performance evaluation, it can be seen that the DS decomposition slightly reduces the variability of the estimation. We also find that while the *grow* and *shrink* supergradients provide comparable performance, the result by the *bar* supergradient is inferior with statistical significance. As seen in Table 1, the *bar* supergradient at $Y$, denoted by $\bar{g}_Y$, is independent on $Y$ indeed, meaning that it does not use local information around $Y$ unlike the *grow* and shrink supergradients. The difference may cause the performance gap of the supergradients. When strict submodularity is introduced with $\varepsilon = 0.001$, we find a slight performance improvement compared to $\varepsilon = 0$, though it is not statistically significant, and the variance is also reduced. In addition, we report the results with the Softplus activation function in Appendix B.

**Set Retrieval** Finally, we consider a set retrieval task for the qualitative evaluation of the DBD. The set retrieval is a task similar to the image retrieval task, where each instance is a set. In this task, we choose an arbitrary query set $X_Q \in \mathcal{D}$. For this query set $X_Q$, we seek top-$K$ similar sets $(\tilde{X}_1, \ldots, \tilde{X}_K)$ that minimizes $D_f(X_Q, X_k)$. If the DBD is learned as a reasonable divergence, the resulting top-$K$ sets should be qualitatively similar to the query set. In this experiment, we set $K = 5$ and query sets are chosen from the airplane and chair classes. Figure 2 shows the experimental results of set retrieval with $\varepsilon = 0$. The leftmost column shows the query set and the right columns are the corresponding top-5 similar sets. In the visualization of point cloud data, the rotation of the

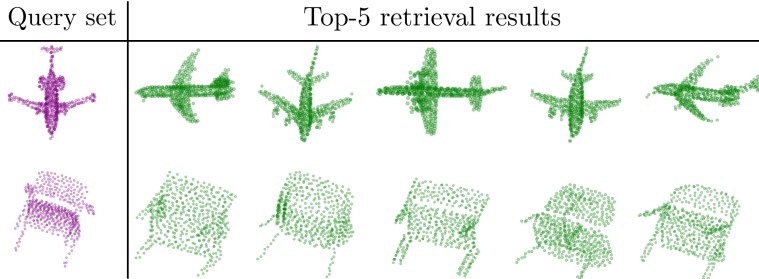

Figure 2: Experimental results of set retrieval for ModelNet40 dataset (Wu et al., 2015). For each given query set, retrieved top-5 similar sets are listed based on the DBD.

camera is arbitrary, but we rotate it for the sake of clarity so that the class to which it belongs is easily recognizable. In addition, since our DBD quantifies the differences between discrete data, it is naturally invariant to these rotations. From the results of this experiment, shown in Fig. 2, it can be seen that the DBD is capable of retrieving similar sets. In particular, the query set and the corresponding similar sets are consistent in the classes to which they belong, indicating that the learning of the DBD is achieved as expected. Also, although omitted in this figure, the divergence between identical query sets $D_f(X_Q, X_Q)$ satisfies $D_f(X_Q, X_Q) < D_f(X_Q, X_k)$, $\forall 1 \leq k \leq n$ and behaves as expected of the divergence, taking the smallest value between identical instances. Additionally, we show the quantitative score on the set retrieval experiment in Table 4 (in Appendix B). Despite using only a simple MLP architecture without any pretraining, our method closely approaches the state-of-the-art method (Hamdi et al., 2021) and achieves better performance than its previous method (Liu et al., 2019). This clearly demonstrates the capability of the DBDs.

## 6 CONCLUSION AND DISCUSSION

Traditional Bregman divergences are defined over continuous spaces using convex functions. In this study, we extended this concept to discrete finite sets by proposing a difference-of-submodular Bregman divergence. This extension enables natural handling of data structures such as point clouds and item sets.

While previous approaches have used submodular functions—analogous to convex functions—to define submodular Bregman divergences on discrete sets, we showed that difference-of-submodular Bregman divergences can be defined from any set function, not necessarily submodular, by utilizing difference-of-submodular decomposition. Theoretically, we showed that enhancing the expressive power of the set function leads to a more expressive difference-of-submodular Bregman divergence. This finding motivates the construction of difference-of-submodular Bregman divergences by learning flexible set functions for specific tasks.

To achieve this, we proposed an approach that learns set functions for difference-of-submodular Bregman divergences using permutation-invariant neural networks, particularly $\varepsilon$-PointNet. Experimentally, we confirmed that the difference-of-submodular Bregman divergence generated from a set function defined as the difference of set functions represented by $\varepsilon$-PointNet is a proper divergence. Furthermore, in a clustering task, the PointNet-based difference-of-submodular Bregman divergence—learned from a small amount of ground-truth cluster data—significantly outperformed existing submodular Bregman divergences constructed on fixed submodular functions.

For future work, the exploration of architectures for the permutation invariant NNs is an important task. We used the simple $\varepsilon$-PointNet architecture in the numerical experiments, but optimal hyperparameters (e.g., number of units and layers, dimension $K$, and $\varepsilon$) were not discussed. Furthermore, novel flexible NN architectures that model submodular functions are proposed (Bilmes & Bai, 2017; Bhatt et al., 2024). From an engineering perspective, it is important to develop a method that performs as well as or better than the state-of-the-art methods according to our framework.

ACKNOWLEDGEMENTS

We thank anonymous reviewers for insightful comments and suggestions. Part of this work is supported by JST CREST Grant Number JPMJCR2015, JST PRESTO Grant Number JPMJPR24K5, JST the establishment of university fellowships towards the creation of science technology innovation Grant Number JPMJFS2136, and JSPS KAKENHI Grant Numbers JP22H03653, 19K20212 and 24K21315.

REPRODUCIBILITY

We provide the code needed to reproduce all experiments in the supplementary material attached. The proofs omitted from the main text are presented in Appendix A.

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

## A  PROOFS

### A.1  WELL-DEFINEDNESS OF STRICT SEMIDIFFERENTIALS

In this section, we show the well-definedness of strict semidifferentials defined in (4) and (5). First, we confirm the equivalence between the strict submodularity and the strict diminished returns property as in (Nemhauser et al., 1978). Hereafter, we denote the increase of a set function $f$ as $f(j|S) := f(S \cup \{j\}) - f(S)$ for $j \in V$ and $S \subseteq V \setminus \{j\}$ and distinguish $\subset$ from $\subseteq$ by whether it contains an equality or not. We repeat the definition of strict submodularity for the sake of convenience:

**Definition A.1.** A set function $f : 2^V \to \mathbb{R}$ is strictly submodular if

$$f(X) + f(Y) > f(X \cup Y) + f(X \cap Y) \tag{10}$$

for every non-comparable $X, Y \subseteq V$.

**Lemma A.2.** *For a set function $f : 2^V \to \mathbb{R}$, the strict submodularity (10) holds if and only if*

$$f(j|S) > f(j|T) \tag{11}$$

*is satisfied for all $S \subset T \subset V$ and $j \in V \setminus T$.*

*Proof.*
[ (10) $\to$ (11) ]
By taking $S \subset T, j \notin T, X = S \cup \{j\}, Y = T$, we have

$$f(S \cup \{j\}) + f(T) > f(S \cup \{j\} \cup T) + f((S \cup \{j\}) \cap T) = f(T \cup \{j\}) + f(S).$$

[ (11) $\to$ (10) ]
Let $X, Y \subset V$ be any non-comparable subsets of $V$ and denote $X \setminus Y = \{j_1, \ldots, j_r\} \neq \emptyset$. From (11),

$$f(j_i | X \cap Y \cup \{j_1, \ldots, j_{i-1}\}) > f(j_i | Y \cup \{j_1, \ldots, j_{i-1}\})$$

holds. By summing up the above equality, the l.h.s. becomes

$$\sum_{i=1}^{r} f(j_i | X \cap Y \cup \{j_1, \ldots, j_{i-1}\}) = \sum_{i=1}^{r} [f(X \cap Y \cup \{j_1, \ldots, j_i\}) - f(X \cap Y \cup \{j_1, \ldots, j_{i-1}\})]$$

$$= f(X \cap Y \cup \{j_1, \ldots, j_r\}) - f(X \cap Y) = f(X) - f(X \cap Y).$$

and the r.h.s. becomes

$$\sum_{i=1}^{r} f(j_i | Y \cup \{j_1, \ldots, j_{i-1}\}) = \sum_{i=1}^{r} [f(Y \cup \{j_1, \ldots, j_i\}) - f(Y \cup \{j_1, \ldots, j_{i-1}\})]$$

$$= f(Y \cup \{j_1, \ldots, j_r\}) - f(Y) = f(X \cup Y) - f(Y).$$

That results in strict submodularity (10). □

Lemma A.2 states the strict version of the diminishing returns property with respect to strictly submodular functions. We can find the following proposition from Lemma A.2.

**Lemma A.3.** *When $f : 2^V \to \mathbb{R}$ is strictly submodular,*

$$f(T) - f(S) < \sum_{j \in T \setminus S} f(j|S) - \sum_{j \in S \setminus T} f(j|S \cup T \setminus \{j\}) \tag{12}$$

*and*

$$f(T) - f(S) < \sum_{j \in T \setminus S} f(j|S \cap T) - \sum_{j \in S \setminus T} f(j|S \setminus \{j\}) \tag{13}$$

*for every $S, T \subseteq V$ ($S \neq T$).*

*Proof.* We use strict diminishing returns (11) to prove (12). Taking any $S, T \subseteq V$, $(S \neq T)$ and denoting $T \backslash S = \{j_1, \ldots, j_r\}$, $S \backslash T = \{k_1, \ldots, k_q\}$. We have

$$
\begin{aligned}
f(S \cup T) - f(S) &= \sum_{t=1}^{r} [f(S \cup \{j_1, \ldots, j_t\}) - f(S \cup \{j_1, \ldots, j_{t-1}\})] \\
&= \sum_{t=1}^{r} f(j_t | S \cup \{j_1, \ldots, j_{t-1}\}) \\
&\leq \sum_{t=1}^{r} f(j_t | S) = \sum_{j \in T \backslash S} f(j | S)
\end{aligned}
\tag{14}
$$

from (11) (the equality holds only if $T \backslash S = \emptyset$). Then,

$$
\begin{aligned}
f(S \cup T) - f(T) &= \sum_{t=1}^{q} [f(T \cup \{k_1, \ldots, k_t\} - f(T \cup \{k_1, \ldots, k_{t-1}\})] \\
&= \sum_{t=1}^{q} f(k_t | T \cup \{k_1, \ldots, k_{t-1}\}) \\
&= \sum_{t=1}^{q} f(k_t | T \cup \{k_1, \ldots, k_t\} \backslash \{k_t\}) \\
&\geq \sum_{t=1}^{q} f(k_t | S \cup T \backslash \{k_t\}) = \sum_{j \in S \backslash T} f(j | S \cup T \backslash \{j\})
\end{aligned}
\tag{15}
$$

is also obtained (the equality holds only if $S \backslash T = \emptyset$). By subtracting (15) from (14), we can confirm (12). Note that the inequality in (12) does not include an equality unlike (14) and (15) because at least one of $T \backslash S$ and $S \backslash T$ is non-empty by the condition $S \neq T$.

Inequality (13) can be derived in a similar manner. We can obtain

$$
\begin{aligned}
f(T) - f(S \cap T) &= \sum_{t=1}^{r} [f((S \cap T) \cup \{j_1, \ldots, j_t\}) - f((S \cap T) \cup \{j_1, \ldots, j_{t-1}\})] \\
&= \sum_{t=1}^{r} f(j_t | (S \cap T) \cup \{j_1, \ldots, j_{t-1}\}) \\
&\leq \sum_{t=1}^{r} f(j_t | S \cap T) = \sum_{j \in T \backslash S} f(j | S \cap T)
\end{aligned}
\tag{16}
$$

and

$$
\begin{aligned}
f(S) - f(S \cap T) &= \sum_{t=1}^{q} [f((S \cap T) \cup \{k_1, \ldots, k_t\} - f((S \cap T) \cup \{k_1, \ldots, k_{t-1}\})] \\
&= \sum_{t=1}^{q} f(k_t | (S \cap T) \cup \{k_1, \ldots, k_{t-1}\}) \\
&= \sum_{t=1}^{q} f(k_t | (S \cap T) \cup \{k_1, \ldots, k_t\} \backslash \{k_t\}) \\
&\geq \sum_{t=1}^{q} f(k_t | ((S \cap T) \cup \{k_1, \ldots, k_q\}) \backslash \{k_t\}) \\
&= \sum_{t=1}^{q} f(k_t | S \backslash \{k_t\}) = \sum_{j \in S \backslash T} f(j | S \backslash \{j\}),
\end{aligned}
\tag{17}
$$

where the equalities in (16) and (17) hold only if $T \backslash S = \emptyset$ and $S \backslash T = \emptyset$, respectively. Inequality (13) is obtained from (16) and (17). $\qquad \square$

We may also be able to show the sufficiency of (12) and (13) for the strict submodularity (10) in a similar way to (Nemhauser et al., 1978, Proposition 2.1), but we omit it because it is not necessary for our methodology.

Now we show the existence of strictly subgradients of strict submodular functions.

**Proposition A.4** (Existence of strict subgradients). *Suppose that* $f : 2^V \to \mathbb{R}$ *satisfies the strict submodularity* (2). *Then, the strict subdifferential of* $f$ *at* $Y \subseteq V$ *is non-empty.*

*Proof.* We show the existence by directly composing specific examples of the subgradients as in (Stobbe, 2013, Section 3.3.1). Let $\lambda \in \mathbb{R}^N$ be some modular function and $t \in \mathbb{R}$ be some real value. We define a modular function $h_Y \in \mathbb{R}^N$ as

$$h_Y := \lambda + t(1_Y - 1_{V \setminus Y}).$$

For any $X \subseteq V$, we can find

$$h_Y(X) - h_Y(Y) = \langle h_Y, 1_X - 1_Y \rangle = \langle \lambda, 1_X - 1_Y \rangle + t \langle 1_Y - 1_{V \setminus Y}, 1_X - 1_Y \rangle$$

and the rightmost term can be rewritten as

$$\begin{aligned}
\langle 1_Y - 1_{V \setminus Y}, 1_X - 1_Y \rangle &= 2 \langle 1_Y, 1_X - 1_Y \rangle - \langle 1_Y + 1_{V \setminus Y}, 1_X - 1_Y \rangle \\
&= 2|X \cap Y| - 2|Y| - |X| + |Y| \\
&= -(|X| + |Y| - 2|X \cap Y|).
\end{aligned}$$

Since the r.h.s. is the negative of the size of the set of $X$ and $Y$ XORed, it takes a negative value when $X \neq Y$. Therefore, we can have

$$\forall X \in 2^V \setminus \{Y\}, \ h_Y(X) - h_Y(Y) = \langle \lambda, 1_X - 1_Y \rangle + t \langle 1_Y - 1_{V \setminus Y}, 1_X - 1_Y \rangle < f(X) - f(Y)$$

by taking sufficiently large $t > 0$. To be precise, we obtain

$$h_Y = \lambda + t(1_Y - 1_{V \setminus Y}) \in \tilde{\partial}_f(Y) \ \Leftrightarrow \ t > \max_{X \in 2^V \setminus \{Y\}} \frac{f(Y) - f(X) + \lambda(X) - \lambda(Y)}{|X| + |Y| - 2|X \cap Y|}.$$

$\square$

Next, let us prove the non-emptiness of strict superdifferentials.

**Proposition 2.5** (Strict supergradients). *The modular functions* $\hat{g}_Y, \check{g}_Y$, *and* $\bar{g}_Y$ *defined in Table 1 are all the strict supergradients if* $f$ *is strictly supermodular.*

*Proof.* For all $X \subseteq V$, we have

$$\hat{g}_Y(X) = \sum_{j \in X} \hat{g}_Y(j) = \sum_{j \in X \setminus Y} f(j|Y) + \sum_{j \in X \cap Y} f(j|V \setminus \{j\}),$$

$$\hat{g}_Y(Y) = \sum_{j \in Y} \hat{g}_Y(j) = \sum_{j \in Y \setminus X} f(j|V \setminus \{j\}) + \sum_{j \in X \cap Y} f(j|V \setminus \{j\}),$$

by definition. If $X \neq Y$, it can be found that

$$\begin{aligned}
\hat{g}_Y(X) - \hat{g}_Y(Y) &= \sum_{j \in X \setminus Y} f(j|Y) - \sum_{j \in Y \setminus X} f(j|V \setminus \{j\}) \\
&\geq \sum_{j \in X \setminus Y} f(j|Y) - \sum_{j \in Y \setminus X} f(j|X \cup Y \setminus \{j\}) \quad (\because \text{Lemma A.2}) \\
&> f(X) - f(Y) \quad (\because \text{Lemma A.3}).
\end{aligned}$$

holds. This indicates $\hat{g}_Y \in \tilde{\partial}^f(Y)$. Similarly, we have

$$
\begin{aligned}
\bar{g}_Y(X) - \bar{g}_Y(Y) &= \sum_{j \in X \setminus Y} f(j|\emptyset) - \sum_{j \in Y \setminus X} f(j|V \setminus \{j\}) \\
&\geq \underbrace{\sum_{j \in X \setminus Y} f(j|\emptyset) - \sum_{j \in Y \setminus X} f(j|Y \setminus \{j\})}_{= \check{g}_Y(X) - \check{g}_Y(Y)} \quad (\because \text{Lemma A.2}) \\
&\geq \sum_{j \in X \setminus Y} f(j|X \cap Y) - \sum_{j \in Y \setminus X} f(j|Y \setminus \{j\}) \quad (\because \text{Lemma A.2}) \\
&> f(X) - f(Y) \quad (\because \text{Lemma A.3})
\end{aligned}
$$

for $X \neq Y$. Thus $\check{g}_Y, \bar{g}_Y \in \tilde{\partial}^f(Y)$ is also obtained. $\qquad\square$

## A.2 STRICT SUBMODULARITY OF THE RELAXED FACILITY LOCATION FUNCTION

**Proposition A.5.** *The relaxed facility location function $f_{\mathrm{FC},\varepsilon}$ defined in* (3) *is strictly submodular for $\varepsilon > 0$.*

*Proof.* For every $X \subset V$ and every $j \in V \setminus X$, we have

$$
\begin{aligned}
f_{\mathrm{FC},\varepsilon}(j|X) &= \varepsilon \sum_{k=1}^{K} \log \left( \sum_{i \in X} e^{\phi_{ik}/\varepsilon} + e^{\phi_{jk}/\varepsilon} \right) - \varepsilon \sum_{k=1}^{K} \log \sum_{i \in X} e^{\phi_{ik}/\varepsilon} \\
&= \varepsilon \sum_{k=1}^{K} \log \left( 1 + \frac{e^{\phi_{jk}/\varepsilon}}{\sum_{i \in X} e^{\phi_{ik}/\varepsilon}} \right). \quad\quad\quad (18)
\end{aligned}
$$

Now, consider $X, Y \subset V$ such that $X \subset Y \subset V$ and $j \in V \setminus Y$. Then,

$$
\begin{aligned}
f_{\mathrm{FC},\varepsilon}(j|X) &= \varepsilon \sum_{k=1}^{K} \log \left( 1 + \frac{e^{\phi_{jk}/\varepsilon}}{\sum_{i \in X} e^{\phi_{ik}/\varepsilon}} \right) \\
&> \varepsilon \sum_{k=1}^{K} \log \left( 1 + \frac{e^{\phi_{jk}/\varepsilon}}{\sum_{i \in Y} e^{\phi_{ik}/\varepsilon}} \right) \\
&= f_{\mathrm{FC},\varepsilon}(j|Y)
\end{aligned}
$$

holds from (18). By Lemma A.2, the strict submodularity of $f_{\mathrm{FC},\varepsilon}$ is shown. $\qquad\square$

# B ADDITIONAL EXPERIMENTS

## B.1 ANOTHER ACTIVATION FUNCTION

Additionally, we report the results of experiments where the activation function in the final layer of $\varepsilon$-PointNet's MLP was changed from ReLU to Softplus. Table 3 shows the results with $\varepsilon = 0$. No significant difference was observed compared to the results obtained with ReLU, as reported in Table 2.

## B.2 QUANTITATIVE EVALUATION FOR SET RETRIEVAL

Table 4 shows the result of the quantitative score of the set retrieval experiment in Section 5.2. Retrieval mean Average Precision (mAP) is employed as the metric as in (Hamdi et al., 2021). We compare the performance of the proposed DBD with PointNet (see Section 5 for the architecture) with two baseline methods, Densepoint (Liu et al., 2019) and the multi-view transformation network (MVTN) (Hamdi et al., 2021). For MVTN, we report the score shown in the original paper (Hamdi et al., 2021) because we failed to reproduce the result.

Table 3: Experimental results of set clustering for ModelNet40 dataset (Wu et al., 2015) with the softplus activation function. Prefixes *grow*, *shrink*, and *bar* correspond to the supergradients $\hat{g}_Y(j)$, $\check{g}_Y(j)$ and $\bar{g}_Y(j)$ respectively (as shown in Table 1). The means and standard deviations of 10 trials with different random seeds are reported.

| $D_f(X,Y)$ | $f(X)$ | Rand index |
|---|---|---|
| *grow*-DBD (softplus) w/ decomposition | $f^1(X) - f^2(X)$ | $0.7940(\pm0.0089)$ |
| *shrink*-DBD (softplus) w/ decomposition | $f^1(X) - f^2(X)$ | $0.7911(\pm0.0090)$ |
| *bar*-DBD (softplus) w/ decomposition | $f^1(X) - f^2(X)$ | $0.7424(\pm0.0107)$ |
| *grow*-DBD (softplus) w/o decomposition | $f^1(X)$ | $0.7743(\pm0.0095)$ |
| *shrink*-DBD (softplus) w/o decomposition | $f^1(X)$ | $0.7756(\pm0.0094)$ |
| *bar*-DBD (softplus) w/o decomposition | $f^1(X)$ | $0.7295(\pm0.0120)$ |

Table 4: Experimental results of set retrieval for ModelNet40 dataset (Wu et al., 2015). Prefixes *grow*, *shrink*, and *bar* correspond to the supergradients $\hat{g}_Y(j)$, $\check{g}_Y(j)$ and $\bar{g}_Y(j)$ respectively (as shown in Table 1). The means and standard deviations of 10 trials with different random seeds are reported, except $*$. Note that the evaluation result with $*$ is borrowed from the original paper.

| Method | mAP |
|---|---|
| *grow*-DBD w/ decomposition | $90.13(\pm0.75)$ |
| *shrink*-DBD w/ decomposition | $90.20(\pm0.77)$ |
| *bar*-DBD w/ decomposition | $86.09(\pm0.85)$ |
| *grow*-DBD w/o decomposition | $88.12(\pm0.80)$ |
| *shrink*-DBD w/o decomposition | $88.20(\pm0.81)$ |
| *bar*-DBD w/o decomposition | $83.57(\pm0.97)$ |
| Densepoint (Liu et al., 2019) | $89.68(\pm0.88)$ |
| MVTN (Hamdi et al., 2021) | $92.9*$ |

