# OpenReview forum: "Difference-of-submodular Bregman Divergence"
_ICLR.cc/2025/Conference — ICLR 2025 Poster_

### Official Review · Reviewer_J6xx · 2024-10-28

**Soundness:** 3
**Presentation:** 2
**Contribution:** 3
**Rating:** 8
**Confidence:** 3

**Summary:**

The authors propose a novel method to construct Bregman divergences for functions defined on discrete sets. Their method is backed by new theory, and it is learnable and applicable to deep-learning on set data.

Their main contributions are:
1. They provide theoretical justification for the technique of [1] to construct Bregman divergences from strictly submodular generator functions. Namely, they prove that the induced divergence is indeed a divergence.
2. They provide a technique to construct Bregman divergences from *arbitrary* generator set-functions, which need not be submodular. They do so using a submodular analogue of the Difference-of-Convex decomposition.
3. They motivate their construction theoretically by proving that a larger class of generator functions makes for a larger class of induced Bergman divergences.
4. They demonstrate the applicability of their method to deep learning via preliminary experiments. In their experiments they use a neural network based on their construction, which computes discrete Bergman divergences that are learnable.
   - They train their architecture to comptue divergences between sets of MNIST digits, and show through examples that the learned metric makes sense.
   - They demonstrate the advantage of their construction compared to simpler ones (based solely on modular set-functions, or on basic set operations) by performing k-Means clustering on PointNet-40, treating the point clouds as sets and using the computed divergences as a distance measure.
   - They further demonstrate their method in the task of set-retrieval on PointNet-40, showing in several examples that the top 5 retrieved examples indeed belong to the correct class.

[1] Faust, Fauzi, Saunderson (2023) - A Bregman Divergence View on the Difference-of-Convex Algorithm

**Strengths:**

All in all I believe that this paper provides valuable contribution to the community, as it lays the theoretical foundations for using discrete Bregman divergences in practical learning tasks, which is an intriguing approach that is not often discussed in this context.

**Weaknesses:**

While the experiments are rudimentary, the paper offers sound theoretical results and demonstrates their applicability to practical learning tasks.

My main concern is that the paper lacks in terms of clarity, particularly in the experiment section. See details below. Should my concerns be addressed, I would be willing to raise my score.

**Questions:**

## Major comments

1. Crucial details regarding your experiments are missing. For example:
   - Line 359: With that loss function, it seems that the global optimum of zero is trivially attainable. What prevents your method from collapsing to zero?
   - What architectures did you use in practice? (dimensions, numbers of layers etc.)
   - Line 307: How did you compute the subgradients and supergradients of the neural networks a priori?

   The paper could seriously benefit from an additional appendix describing the technical details of your experiments.
2. Line 458: "In addition, since our DBD quantifies the differences between discrete data, it is naturally invariant to these rotations. Note that the rotational invariance in point cloud data corresponds to the permutation invariance in set data and this property is not provided by usual divergences over vectors."
   This argument and reasoning are unclear. Are you saying that the DBD between two point-clouds X,Y is rotation invariant, namely D(X,Y) = D(RX,RY) for all rotations R? If this is the case, it is highly nontrivial and requires proof. Anyway the argument itself requires clarification.
3. Line 20: "outperforming existing methods on tasks such as clustering" is too strong a statement, as you did not compare with existing methods outside the realm of discrete set functions.
4. The continuous analog of the core idea in this work, namely to construct Bregman divergences from arbitrary nonconvex generator functions using the Difference-of-Convex decomposition, was studied in [1]. Their contribution should be acknowledged.

## Minor comments

1. How come bar-DBD with decomposition performed worse than grow- and shrink-DBD without decomposition? Is there an intuitive explanation? Are there any examples where you observed a stronger benefit to non-modular over modular generator functions? This could provide stronger empirical support for your theoretical contribution.
2. Lines 422-424: For clarity, I suggest stating explicitly that you calculated the Rand index between the resulting clustering and the clustering induced by the ground-truth labels.
3. In l.213-218, I suggest replacing "a formal discussion on the well-definedness is lacked", which sounds vague, with a clear and explicit statement of what you prove that the referred paper did not.

## Possible errata

1. Lines 213-218 and 236: The citation (Iyer & Bilmes 2012a) should probably be (Iyer & Bilmes 2012b).
2. Line 447, the first instance of 'bar' should probably be 'shrink'.
3. Line 701: 'and' in "and for every" should probably be removed.

---

> ### Author Response · Authors · 2024-11-22
>
> We thank you for your careful review and comments. We addressed your comments and questions below:
>
> ### Trivial Solution of the Loss Function
> The loss function $L$ is minimized for $f$, not for $(X_A, X_P, X_N)$; therefore, there may be no concern about falling into a trivial solution as suggested by your question. In the submitted version of our paper, we denoted the definition of the triplet loss as $L(X_A, X_P, X_N)$ even we want to minimize the loss with respect to $f$. We have updated the notation to make it less confusing (lines: 360-364). If we have misunderstood the intent of your concern, please let us know. Thank you for your question.
>
> ### Architectures
> Thank you for pointing it out. We have added details about the network architecture on line (lines: 351-359).
>
> ### Taking Semigradients
> For the supergradients, we use grow, shrink, and bar supergradients (Table 1). For the subgradients, we take the extreme point, which is also used in Iyer & Bilmes (2012b). The algorithm to take the subgradient was noted in Section 2.1 (lines: 137-140), but not in the section of the experiment. We have clarified it in Section 5 (lines: 367-369).
> Note that once a sequence of inputs (a sequence of coordinates in the case of point clouds) is given to a permutation-invariant NN $f_{\mathrm{NN}}$, $f_{\mathrm{NN}}$ can be regarded as merely a set function. Then, the subgradient and supergradient are calculated.
>
> ### Rotation Invariance and Permutation Invariance
> The method used to achieve permutation invariance does not guarantee rotational invariance. Therefore, the statement in question is a mistake in our description. Point cloud data is a set of unordered points, where the order of the points is meaningless. As such, it is desirable for a network processing point clouds to produce outputs that are invariant to changes in the input order. By using the PointNet implementation, we have achieved permutation invariance as originally intended. We have removed the sentences. Thank you for your question.
>
> ### Suggested Citation
> We have added the citation to Faust, Fauzi, Saunderson (2023) at Section 1 (lines:95-96). We overlooked that important paper. Thank you for your suggestion.
>
> ### Why is the Result by the Bar-supergradient worse?
> This is a good question. As seen in Table 1, the bar-supergradient $\bar{g}_Y$ is indeed defined independently on $Y$, unlike grow- and shrink-supergradients. Intuitively, this means the bar-supergradients cannot take the local information around $Y$ into account and it causes the performance degradation. We have added the discussion in Section 5.2 (lines: 462-466).
>
> ### Benefit of Using Non-submodular Generating Functions
> Consider $f = f^1 - f^2$ is modular, that is, $f = m$ holds for a modular function $m$. Then, the DBD (Eq. 9) is also modular. In this case, the DBD ultimately reduces to a simple set operation (and its weighting), and as pointed out in the introduction (lines: 46-50), its representational power becomes highly limited. To address this issue, we believe that defining set function divergences via submodular functions provides a fundamental solution.
>
> ### Clarification about the Rand Index
> Thank you for your suggestion. We have clarified that we used class labels as the reference clusters for computing the Rand index (lines: 450-453).
>
> ### Revise the Statement
> > In l.213-218, I suggest replacing "a formal discussion on the well-definedness is lacked", which sounds vague
>
> Thank you for your comment. We agree to your comment and have changed the representation (lines: 229-231).
>
> ### Eratta
> Thank you very much for carefully reviewing and pointing out these details. We have appropriately revised these points.
>
> Again, thank you very much for your comments and questions.

---

> > ### Comment · Reviewer_J6xx · 2024-11-22
> > **Response to Authors**
> >
> > Thank you for addressing all of my concerns. I have increased my score accordingly.

---

> > > ### Author Response · Authors · 2024-11-27
> > >
> > > Thank you for reviewing our rebuttal and updating the scores! We sincerely appreciate your thorough and thoughtful review.

---

### Official Review · Reviewer_JSTp · 2024-10-29

**Soundness:** 2
**Presentation:** 2
**Contribution:** 3
**Rating:** 6
**Confidence:** 3

**Summary:**

Bregman divergence is a pseudo-distance for comparing vectors or functions. In this paper, the authors present a new technique to construct such divergence on a discrete domain. Specifically:

1. The authors prove that a strictly submodular function can induce a Bregman divergence (Theorem 3.1).
2. They extend this property, showing that when the function has the form $f + m$, where $m$ is a modular function and $f$ is neither submodular nor supermodular, it can also induce a Bregman divergence (Theorem 3.1').
3. Finally, they demonstrate that the broader the function class, the broader the class of induced divergences (Theorem 3.4).
4. They provide a numerical form of the proposed discrete Bregman divergence (Section 4).
5. Numerical experiments show that the learnable discrete Bregman divergences can capture structure and outperform existing methods in downstream tasks.

**Strengths:**

1. The authors extend previous work on constructing Bregman divergence by relaxing the requirement that the generator function $f$ must be submodular.
2. They present a numerical form of the proposed discrete Bregman divergence using permutation-invariant neural networks.
3. In the experiment section, the authors demonstrate that the constructed Bregman divergence returns smaller values for similar set pairs and larger values for dissimilar set pairs.

**Weaknesses:**

1. In Section 5.2, *Real Data Application*, it appears that there are no baselines for either the clustering or shape retrieval experiments. Figure 2 demonstrates the performance of the proposed Bregman divergence, but it is difficult to see how this new divergence improves upon previous divergences, such as those presented in [Iyer and Bilmes (2012a)].

Additionally, quantitative metrics (e.g., accuracy in the shape retrieval/classification experiment) and wall-clock comparisons to assess the performance of the proposed divergence should be included and discussed.

2. The differences between the new divergence construction technique and previous work [Iyer and Bilmes (2012a)] remain unclear. For instance, in Equation (7), based on Iyer’s work, $f$ should be submodular, and $h_Y$ serves as its subgradient, which is straightforward to construct.

In the new divergence construction technique proposed here (presumably based on Theorem 3.1'), $f$ can be any set function, which raises the challenge of finding the appropriate modular mapping $h_Y$. In short, the old technique imposes a requirement on $f$, making $h_Y$ straightforward once $f$ is constructed. The new technique has no requirement for $f$ but requires finding an appropriate $h_Y$. Given this, it is not immediately clear why the new technique would outperform the previous one.

**Questions:**

1. Regarding the notation in Theorem 3.1, it appears that $h_Y$ and $g_Y$ are indeed independent of the set $Y$. Is that correct?

2. In Table 2, could you clarify the values in each column? Specifically, are they the mean and variance of 10 trials of what particular measure?

3. In Theorem 3.1' and the contribution section (lines 069-071), the authors mention proving that $f$ does not need to be submodular. However, in Section 4, lines 306-307, it appears that the implementation still requires submodular functions $f_1$ and $f_2$. What, then, are the advantages and differences of the new divergence construction technique compared to the original one by [Iyer and Bilmes (2012a)], which requires submodularity? It seems that the proposed technique still relies on submodular properties in the implementation stage.

---

> ### Author Response · Authors · 2024-11-22
>
> We thank you for your careful review and comments. We addressed your comments and questions in turn.
>
> ### Dependence on $Y$ of semigradients
> The answer to this question is partially yes and partially no. While (strict) subgradients and supergradients generally depend on $Y$, they may also be determined independently of $Y$. In Table 1, the definitions of the shrink supergradient and grow supergradient depend on $Y$. Additionally, for subgradients, the algorithm mentioned in Section 2.1 (lines 137-140) for finding the extreme point also relies on $Y$. On the other hand, the bar supergradient, as defined in Table 1, does not depend on $Y$.
>
> ### Clarity of Table 2
> Thank you for pointing it out. In lines 450-453 we have clarified that the values of the rightmost column in Table 2 indicate the Rand index between the estimated clusters and the true class labels.
>
> ### Difference from Iyer & Bilmes (2012a)
> It is indeed correct that submodularity remains necessary. However, we believe that our introduction of the DS decomposition, which enhances the representational power, is a noteworthy contribution both theoretically (Theorem 3.4) and empirically (Table 2). Moreover, the PointNet model used in our implementation has been shown to possess universal approximation capabilities in the original paper (Qi et al., 2017). In our study, we fixed $\gamma$ in Definition 2.7 to ensure submodularity, but if PointNet retains sufficient representational power under this constraint, our DBD implementation using the two submodular functions $f^1$ and $f^2$ should enable representation of a broad class of set functions $f=f^1 - f^2$. Combined with our discussion on the definability of the (generalized) Bregman divergence for general set functions, we believe our proposed practical construction method offers a significant progress from the prior study.
>
> Again, thank you very much for your comments.

---

> ### Author Response · Authors · 2024-11-27
>
> ### Lack of Baseline Methods in the Set Retrieval Experiment
> As an update feasible within the limited rebuttal period, we conducted a quantitative evaluation on the retrieval task using MVTN (Hamdi et al., 2021), considered SoTA, and its predecessor, DensePoint (Liu et al., 2019). The results have been added to Table 4 (Appendix C). The key findings are as follows:
> - Despite not employing pretraining, complex hyperparameter tuning, or elaborate architecture design, our method outperformed DensePoint and achieved performance close to MVTN. This demonstrates that our DBD-based framework is sufficiently effective in practical scenarios.
> - As in the set clustering task, the quantitative evaluation of the set retrieval task also showed that w/ decomposition ($f = f^1 - f^2$) outperformed w/o decomposition ($f = f^1$, which is effectively equivalent to the submodular Bregman divergence proposed by Iyer & Bilmes (2012b)).
>
> Regarding the comment:
> > Figure 2 demonstrates the performance of the proposed Bregman divergence, but it is difficult to see how this new divergence improves upon previous divergences, such as those presented in [Iyer and Bilmes (2012a)].
>
> We believe that the quantitative evaluations provide clear evidence of the superiority of our DBD over the work by Iyer & Bilmes (2012b).
>
> ### Computational Time
> To be honest, due to the time constraints of the rebuttal period and queueing issues with cloud computing resources, we were unable to measure precise wall-clock time in a unified computational environment.
>
> As a reference, in our representative computational environment (Ubuntu 22.04 with 16 GiB RAM and 3.6 GHz 4 vCPUs, without GPU usage), the approximate computation times for learning the DBDs are as follows:
> - MNIST: About 30 seconds per epoch, totaling approximately 100 minutes for 200 epochs.
> - ModelNet40: About 80-100 seconds per epoch, totaling approximately 4.5-5.5 hours for 200 epochs.
>
> These times include data loading overhead and reflect the lack of GPU acceleration and optimization in our implementation. With proper engineering and the use of GPUs, significant speedups are expected.
>
> ### References
> Abdullah Hamdi, Silvio Giancola, and Bernard Ghanem. MVTN: Multi-view transformation network for 3d shape recognition, ICCV2021.
> Yongcheng Liu, Bin Fan, Gaofeng Meng, Jiwen Lu, Shiming Xiang, and Chunhong Pan. Densepoint: Learning densely contextual representation for efficient point cloud processing, ICCV2019.

---

> > ### Author Response · Authors · 2024-11-29
> >
> > If you have any additional questions or feel that our response remains inadequate in any way, please feel free to let us know. We welcome further discussion throughout the remainder of the discussion period.

---

> > > ### Comment · Reviewer_JSTp · 2024-12-02
> > >
> > > The reviewer thanks the authors for their response. I have raised my score.

---

### Official Review · Reviewer_QZP6 · 2024-11-03

**Soundness:** 2
**Presentation:** 3
**Contribution:** 2
**Rating:** 5
**Confidence:** 3

**Summary:**

In this work, the authors generalize discrete Bregman divergences (DBDs) introduced in Iyer and Bilmes (2012a) to the case where the generating functions are not restricted to be submodulars. By leveraging the fact that any set function can be decomposed as the difference of two submodular ones, the authors show that any set function induces a DBD, and that this extension enables to define larger classes of DBDs. Additionally, they propose a framework to learn such divergences from observations by leveraging existing permutation invariant architectures, such as PointNet, that are by construction submodulars. While obtaining the decomposition of a set function as a difference of two submodular ones take exponential complexity, the authors propose to directly model generalized DBDs using the difference of two parametrized submodular functions obtained from PointNets. Then, they propose to learn an adapted DBD from labeled observations using the triplet loss, and show experimentally the application of their approach for clustering and retrieval tasks on two real-world datasets.

**Strengths:**

The paper is well-written and presented in a clear, accessible manner. The authors thoroughly acknowledge the relevant literature and prior work, thereby enhancing the clarity of their contributions. To the best of my knowledge, the proposed generalization of DBDs is novel, as is the framework introduced for learning them.

**Weaknesses:**

The main weaknesses of the paper are twofold: (1) a lack of motivation, and, relatedly, (2) a lack of empirical evidence to demonstrate the benefits of the proposed approach. Regarding the first point, while the paper introduces a new mathematical tool—the (generalized) DBD—the motivation for its introduction is not sufficiently developed. Although the authors demonstrate that their extension allows the definition of larger classes of DBDs, the motivations for considering DBDs in the first place for ML tasks are not clearly articulated. This should be clarified in the introduction and related work. Concerning the second point, the authors aim to demonstrate the applications of this tool for clustering and retrieval tasks; however, the experimental results might be insufficient to demonstrate how the proposed generalization improves upon standard DBDs using only submodular generating functions, as only one experiment is provided to demonstrate this point. Furthermore, it remains unclear how DBDs compare to other baseline methods capable of addressing similar tasks as no comparative analysis has been carried out. To enhance the impact of their work, I suggest that the authors strengthen the motivation, and compare the proposed approach with SoTA baselines on the tasks considered.

**Questions:**

How does the proposed approach compare with other baselines for clustering and retrieval tasks?
Can the authors provide more experimental evidence on the advantage of considering their generalized DBD rather than using the standard one with submodular generating functions?

**Details Of Ethics Concerns:**

N.A

---

> ### Author Response · Authors · 2024-11-22
>
> We thank you for your careful review and time. We addressed your comments and questions as below:
>
> ### Lack of Motivation
> In the introduction, we have presented numerous examples of how Bregman divergences appear in machine learning and statistics, particularly in continuous spaces. Given their foundational importance in these fields, we believe their significance is widely recognized and undisputed. Building on this established foundation, we believe that our attempt to apply Bregman divergences to discrete spaces and to consider its generalization with the practical meaningfulness is a natural and well-motivated approach. We hope this perspective clarifies the motivation and relevance of our approach.
>
> Again, thank you very much for your review.

---

> > ### Comment · Reviewer_QZP6 · 2024-11-26
> >
> > Thank you for the response. My concerns have not been fully addressed, so I will be maintaining my current score.

---

> > > ### Author Response · Authors · 2024-11-27
> > >
> > > Thank you for your prompt reply.
> > >
> > > ### Lack of Empirical Evidence
> > > Regarding the motivation, it has already been addressed in the rebuttal we submitted earlier, so we assume the unresolved concern lies with the baselines. Since there were no specific comments on the selection of comparative methods, we conducted experiments within the limited time available. We chose methods that we considered both reasonable and feasible for the retrieval task, focusing on SoTA methods applicable to the task: MVTN (Hamdi et al., 2021) and its predecessor, DensePoint (Liu et al., 2019). The results have been added to Table 4 (Appendix C).
> > > The findings are as follows:
> > > - Despite not employing pretraining, complex hyperparameter tuning, or elaborate architecture design, our method outperformed DensePoint and achieved performance close to MVTN. This suggests the effectiveness of our DBD-based framework.
> > > - As with the quantitative evaluation of set clustering, w/ decomposition ($f = f^1 - f^2$) outperformed w/o decomposition ($f = f^1$, which is effectively equivalent to the submodular Bregman divergence by Iyer & Bilmes (2012b)). This provides substantial evidence that generalizing the submodular Bregman divergence was beneficial.
> > >
> > > We hope these additional results help address your concerns, at least partially. If further clarification is needed, we are open to providing additional explanations or experiments within the remaining discussion period.
> > >
> > > ### References
> > > Abdullah Hamdi, Silvio Giancola, and Bernard Ghanem. MVTN: Multi-view transformation network for 3d shape recognition, ICCV2021.
> > > Yongcheng Liu, Bin Fan, Gaofeng Meng, Jiwen Lu, Shiming Xiang, and Chunhong Pan. Densepoint: Learning densely contextual representation for efficient point cloud processing, ICCV2019.

---

> > > > ### Author Response · Authors · 2024-11-29
> > > >
> > > > If there are any remaining concerns or if our response is still insufficient in any way, please do not hesitate to let us know. We welcome continued discussion!

---

> > > > > ### Comment · Reviewer_QZP6 · 2024-11-30
> > > > >
> > > > > Thank you for providing additional experiments. Although the authors have added baselines, I still consider that the paper lacks empirical evidence and motivation to justify the introduction of their generalization. However, I note the authors' effort to improve the points I raised, and so I decide to increase my score.

---

### Official Review · Reviewer_VMyv · 2024-11-04

**Soundness:** 2
**Presentation:** 3
**Contribution:** 2
**Rating:** 8
**Confidence:** 2

**Summary:**

The submission proposes new class of divergences on discrete spaces and a learning framework comprised of permutation-invariant neural networks with metric learning loss.

Leveraging difference-of-submodular (DS) decomposition for any set function f, the authors obtain more expressive class of Bregman divergences dubbed discrete Bregman divergences (DBD). Expressiveness advantage over submodular Bregman divergences is achieved by extending the underlying set function class to not necessarily be submodular but rather admit DS decomposition.

The paper validates the proposed approach to learning DBD in a set of numerical experiments.

**Strengths:**

The paper addresses a problem of extending Bregman divergences to discrete spaces. This is an important problem as many methods that rely on Bregman divergences in continuous settings may be adapted to discrete scenarios.

The submission does a good job providing a structured and clear background on the treated problem.

The central idea of the paper stems from DS decomposition. Typically, one needs a submodular function $f$ to instantiate submoduler Bregman divergence. However, the authors notice that DS decomposition proposed in prior work alleviates this constraint on $f$ itself, and propose to use the difference of two submodular functions, $f = f^1 - f^2$. I'm not an expert in the area, but it seems the idea of using DS decomposition in forming more expressive class of Bregman divergences on discrete spaces has not been explored in the literature and is novel.

The authors provide empirical validation for the proposed learning framework. Given two submodular $f^1, f^2 $ functions, DS decomposition facilitates discrete Bregman divergence specification and with proper implementation of $f^1, f^2$ one can utilize metric learning approach to learn divergence from data. This defines a natural combination of permutation-invariant neural networks (to work with sets) and triplet loss (to learn the divergence). With MNIST experiment, they illustrate that the learned divergence provides reasonable results for dissimilarity between sets of images, with similar image sets getting smaller dissimilarity scores. Point cloud dataset experiments validate that the use of novel divergence class provides benefits over the use of submodular Bregman divergences in set clustering experiment. They further validate that the divergence learned provides semantically close set retrieval.

**Weaknesses:**

At the very start, the paper highlights the problem of identifiability of divergence, which comes from divergence definition $D(x, y) = 0 \rightarrow x=y$. Thus, later the submission puts a lot of attention on the strict submodularity of the underlying set functions in Bregman divergences as one needs strict inequalities to satisfy the definition requirement. However, the proposed implementation seems problematic to me in this sense. Indeed, the submodularity is guaranteed, but DBD requires strict DS decomposition, so $f^i$s should be strictly submodular, which doesn't seem the case for the adopted architecture:
$$
f_{PN}([x_i]_{i \in X}) = \max_{i \in X} ReLU(h(x_i)),
$$
where $h: \mathcal{X} \rightarrow \mathbb{R}$ instantiated with an MLP with last activation set to ReLU. When argmax is achieved outside the intersection, the inequality from Definition 2.3 holds, but if argmax is inside the intersection, we fail to meet strict modularity. So the proposed implementation doesn't match the requirements of DBD. Perhaps I'm missing something here?

Since the proposed approach targets practical side of things, it seems it needs more extensive experimentation. For example, one would expect to see expressiveness comparison that is not limited to only one task (set clustering) and one dataset (ModelNet40). This will help gain more empirical evidence for the substantial gain in expressiveness across tasks and task complexities, and justify the use of DBD which requires more computational resources (two functions instead of one).

The authors didn't discuss the limitations of their approach, which partly stems from limited experimentation.

**Questions:**

Formally, the proposed implementation is not divergence as per the Definition 1.1. But rather the generalized divergence? What are the consequences of that in practice? In what scenarios the difference may play a crucial role?

As noted in the weaknesses section, the expressiveness of the new DBD is fully explored empirically. Can we quantify how much more expressive the new class of divergences is compared to the submodular Bregman divergences? I think this is important to show the strength of the new DBD and should be explored empirically.

---

> ### Author Response · Authors · 2024-11-22
>
> Firstly, we thank you for your careful review and comments. We addressed your comments and questions in turn.
>
> ### Non-strict Submodularity of PointNet
> This is a very essential question. Your concern is valid; a vanilla PointNet with ReLU in the final layer of the MLP does not satisfy strict submodularity. Initially, we used the vanilla PointNet for simplicity of explanation (and due to space constraints), but we agree that this may have caused some confusion. We are currently conducting experiments with the activation function changed to Softplus, which addresses the concern about strict submodularity.
>
> ### Expressiveness of the DBD
> > the expressiveness of the new DBD is fully explored empirically
>
> In response, we would like to clarify that this statement is not accurate. In Theorem 3.4, we demonstrate that the proposed DBD truly has superior representational power compared to the conventional submodular-Bregman divergence. We believe this theorem will likely address your concerns, so please have a look.
>
> Again, thank you very much for your comments.

---

> ### Author Response · Authors · 2024-11-27
>
> ### Non-strict Submodularity of PointNet (Contd.)
> Initially, we attempted to address the issue of non-strict submodularity using the Softplus activation function, but we realized this was a misguided approach. Ultimately, we sought a fundamental solution to this issue through the following steps:
> 1. We defined an $\varepsilon$-PointNet architecture by replacing the max function in PointNet with the logsumexp function (i.e., $f_\varepsilon(\boldsymbol{x}) = \varepsilon \log \sum_i \exp (x_i / \varepsilon)$)⁡, which converges to vanilla PointNet in the limit as $\varepsilon \to 0$.
> 2. We proved that the $\varepsilon$-PointNet allows for the realization of a permutation-invariant NN that guarantees strict submodularity (Appendix B).
> 3. In the set clustering experiment, we added results for $\varepsilon=0$ (i.e., vanilla PointNet) and $\varepsilon=0.001$ (strictly submodular), showing experimentally that the latter achieves equal or better performance (Table 2).
>
> Due to time constraints, we were unable to conduct experiments with $\varepsilon-PointNet$ on tasks other than set clustering. However, we believe that this update fundamentally addresses your concerns. Additionally, we added the set clustering results with the Softplus activation functions in Appendix C.
>
> ### Expanding Experiments
> Due to time constraints, we were unable to add experiments on new datasets or tasks. As a feasible alternative, we conducted additional baseline comparisons and quantitative evaluations on the retrieval task of the ModelNet40 dataset. The results show that our method achieves scores close to those of SoTA methods, despite not involving pretraining, complex hyperparameter tuning, or elaborate architecture design. We believe this reflects the practical strength of our DBD framework.
>
> Furthermore, regarding your comment:
> > This will help gain more empirical evidence for the substantial gain in expressiveness across tasks and task complexities, and justify the use of DBD which requires more computational resources (two functions instead of one).
>
> We would like to point out that the results labeled w/o decomposition in the tables correspond to the case where only a single function was used. Notably, as described in Section 5, "Network Architecture," despite w/o decomposition ($f = f^1$) having more weight parameters than w/ decomposition ($f = f^1 - f^2$), the quantitative evaluation results demonstrate that w/ decomposition achieves superior performance. We hope this clarifies your concerns.

---

> > ### Comment · Reviewer_VMyv · 2024-11-27
> > **Thank you!**
> >
> > Dear authors, thank you for addressing the concerns raised in the review. Although larger-scale empirical validation has not been added due to time limitation, the authors still managed to add quantitive evaluation and comparison to the baseline methods on the considered dataset. I have raised my score accordingly.

---

> > > ### Author Response · Authors · 2024-11-27
> > >
> > > Thank you for promptly reviewing the revision and for improving the score! We sincerely appreciate your thorough and thoughtful review.

---

### Official Review · Reviewer_d1i9 · 2024-11-09

**Soundness:** 4
**Presentation:** 3
**Contribution:** 4
**Rating:** 8
**Confidence:** 5

**Summary:**

Review of "DISCRETE BREGMAN DIVERGENCE" submitted to ICLR 2025.

This paper extends the submodular Bregman divergence methods introduced a few years ago to non-submodular functions. They do this via the notion of a DS (difference of submodular) decomposition of non-submodular functions. The key thing is that in order for the identifiability property of the Bregman divergence to hold (i.e., that D(x,y) = 0 iff x==y), they note that the submodular function needs to be strict (i.e., lie on the interior of the submodular cone), something they point out was not mentioned in the past (although arguably it was implicit). They note that any set function has a DS decomposition in terms of two strict submodular (or two strict supermodular) functions and since submodular functions have both sub-gradients and super-gradients, it is possible to use the two DS components of any set function to define a sub-gradient and a super-gradient based Bregman divergence, and therefore define a Bregman divergence for any set function.

They go on to show that these functions can be learnt, i.e., one can learn two submodular functions and then find the semi-gradients of these to produce a discrete Bregman divergence based on these learnt submodular functions. They show results on some set clustering that seem to me to be reasonable.

While I do not think that the paper is revolutionary, and I do think that the strictness of previous DS decompositions was implicit, I do think it is worth pointing that out more explicitly as this paper (and also the Li & Du, 2020) in fact do, so I agree with this. There are a few issues of tone, however, that I would change in the paper and that I point out below. Also there are a few recent citations that I think you should add. All in all, however, I think the paper should be accepted as it is a nice contribution, and it is in particular good to see the empirical work using their submodular-supermodular Bregman divergence methods.

Here are some comments.

Firstly, I think you might consider changing the title of the paper a Submodular-Supermodular Bregman Divergence, or "Discrete DS Bregman Divergence" since the approach is entirely dependent on there being a DS decomposition. If one only has oracle access to a non-submodular non-supermodular set functions, it can be hard to find a reasonable decomposition (assuming one knows bounds of the function, one can always add and subtract a very large strict submodular function to any set function to transform it to a DS function but that is a fairly vacuous DS decomposition). So unless you really can produce a Bregman Divergence for any set function given only oracle access, I think it is more appropriate to entitle your paper "Submodular-Supermodular Bregman Divergence".

I think you may want to change the tone of lines 213-216 where you say "a formal discussion on the well-definedness is lacked" as that sounds a bit disparaging. You are basing your results strongly on their methods, standing on their shoulders, so you might say something along the lines of "This earlier work, however, did not explicitly mention that in order for the identifiability property of the Bregman divergence to always hold, it is necessary for the submodular functions involved to be strict", i.e., be more explicit in what you are building on rather than saying that the previous paper "is lacked".

I think the numerical experiments are good and, as mentioned above, it is good to see empirical work as well on submodularity, I think there should be more such things.

The submodular functions that you are learning however seem to be either deep submodular functions (DSFs), i.e., see (Bilmes & Bai from 2017, https://arxiv.org/abs/1701.08939) or much more recently deep submodular peripteral networks (DSPNs, https://arxiv.org/abs/2403.08199 from 2024). I think both papers should be cited. In particular, it seems your submodular functions are simple forms of DSPNs, but I think that one could learn two DSPNs and construct one of your Bregman divergences from semi-gradients of DSPNs quite easily, and this would both further extend the expressivity of your Bregman divergences and also extend the utility of these DSPNs. Also DSPNs strictly extend DSFs (removing the only known limitation of DSFs), and this is also useful for discrete Bregman divergences.

**Strengths:**

see above.

**Weaknesses:**

see above.

**Questions:**

see above.

---

> ### Author Response · Authors · 2024-11-22
>
> We thank you for your insightful feedback and time. We addressed your requests as below:
>
> ### Change the Title of the Paper
> We changed the title of the paper from "Discrete Bregman Divergence" to "Difference-of-submodular Bregman Divergence." The revised title slightly differs from your suggestions, but we believe our changes are in line with your intention.
>
> ### Revision of a Sentence
> > I think you may want to change the tone of lines 213-216 where you say "a formal discussion on the well-definedness is lacked" as that sounds a bit disparaging.
>
> We can completely agree with your opinion. We changed the representation according to your suggestion (lines: 229-231). Thank you for pointing it out.
>
> ### Add Citations
> Thank you for pointing out the lack of related papers that should be mentioned. We added a paragraph at the end of Section 6 and cited the DSF and DSPN papers there.
>
> Again, thank you very much for your comments.

---

### Meta-Review · Area_Chair_V7Bx · 2024-12-18

**Metareview:**

This paper extends Bregman divergence, typically derived from convex functions, to cases where the generating function is neither submodular nor supermodular, resulting in the difference-of-submodular Bregman divergence. This further extends the framework introduced by Iyer & Bilmes (2012b) which used submodular functions as generating functions.  A learnable version of this divergence is proposed using permutation-invariant neural networks, showing through experiments that it captures key structural properties in discrete data and improves performance in clustering and set retrieval tasks.

Reviewers generally agree that the paper provides a new and useful tool for tasks requiring structure-preserving distance measures, and achieving it requires overcoming a few nontrivial obstacles with new insights and techniques.  That said, the empirical evaluation can still be much improved.  Overall, the paper forms an interesting addition to the conference.

**Additional Comments On Reviewer Discussion:**

The rebuttal has been noted by the reviewers and have been taken into account by the AC in the recommendation of acceptance/rejection.

---

### Decision · Program_Chairs · 2025-01-22

Accept (Poster)